# An experimental Study Investigating the Effects on *Brassica oleracea*: Estuarine Seaweeds as Biostimulants in Seedling Development?

Madalena Mendes [1], Diana Pacheco [1], João Cotas [1], Kiril Bahcevandziev [2,3] and Leonel Pereira [1,*]

1 Department of Life Sciences, MARE-Marine and Environmental Sciences Centre, University of Coimbra, 3001-456 Coimbra, Portugal
2 Polytechnic Institute of Coimbra, Coimbra Agriculture School, Bencanta, 3045-601 Coimbra, Portugal
3 Research Centre for Natural Resources, Environment and Society (CERNAS), Coimbra Agriculture School, Bencanta, 3045-601 Coimbra, Portugal
* Correspondence: leonel.pereira@uc.pt; Tel.: +351-239-240-782

**Abstract:** Estuarine eutrophication due to the nutrient run-off from the agricultural fields encourages the establishment of several opportunistic seaweeds. These fast-growing seaweeds, considered an untapped resource, with a circular economy approach, can be employed as soil plant fertilizer. In agriculture, there is a global trend toward shifting from chemical-based conventional farming to sustainable agriculture. In this context, this study aimed to understand the biostimulant potential of the aqueous extracts from seaweeds harvested in Mondego estuary located on the Atlantic coast of Portugal, namely *Ulva lactuca* (Sea lettuce), *Fucus ceranoides* (Estuary Wrack) and *Gracilaria gracilis* (Slender Wart Weed), in kale (*Brassica oleracea* L.) seed germination and seedling development. The results showed that *Gracilaria gracilis* extract enhanced kale seed germination, presenting a higher seedlings weight ($0.076 \pm 0.004$ g) and length ($15.48 \pm 0.59$ cm), when compared with seedlings obtained in distilled water used as a control (weight = $0.059 \pm 0.002$ g; length = $13.10 \pm 0.54$ cm). *Fucus ceranoides* showed the lowest influence on seedling development (weight = $0.062 \pm 0.002$ g; length = $12.97 \pm 0.59$ cm). However, these results demonstrated that seaweed aqueous extracts can indeed enhance seed germination and seedling development.

**Keywords:** seaweed aqueous extracts; biostimulant; seedling development; *Brassica* spp.

## 1. Introduction

Considering the rapidly increasing world population and the impact of the occurrence of biotic and abiotic stresses in food production, it is important to apply novel and more efficient techniques to improve agriculture yield in a sustainable way.

Kale or leaf cabbage (*Brassica oleracea* L.) is a vegetable that belongs to the Brassicaceae family, and it is a globally cultivated food source [1]. According to Food and Agriculture Organization (FAO) data, Europe is, after China, the second biggest brassicas producer in the world (average of 52,630,430 tons in the last 10 years) and in recent years, there has been a subtle decrease in the ratio production/harvested area (yield) due to inefficient and excessive use of inorganic fertilizers, which are agricultural systems recognized to be a significant source of environmental damage [2,3].

The traditional approach in the agriculture sector has a huge environmental impact, being one of the causes of high levels of residual nitrate in the soil due to excessive use of nitrogen (N) fertilizers, causing soil acidification, river water pollution (consequently, groundwater) and high $N_2O$ emissions released from soils [4]. More than 60% of global $N_2O$ emissions come from agricultural soils, with bacterial and fungal denitrification and nitrification processes accounting for more than two-thirds. $N_2O$ is generated in ammonia-oxidizing bacteria by the oxidation of hydroxylamine to nitrite. Nitrate is converted to $N_2$

through denitrification, a sequential reduction of $NO_3^-$ to $NO_2^-$, NO, $N_2O$ and finally, $N_2$, [5]. In addition to denitrification, another significant nitrate-reducing mechanism in soil is respiratory nitrate ammonification, responsible for the loss of nitrate and the generation of $N_2O$ from the reduction of NO that is created as a by-product of the reduction process [6]. All of the above contribute to increased emissions of global greenhouse gases (GHGs) and make the soil less useable for crop production [3].

Seaweeds have enormous potential for reducing global warming and climate change. Some cultivated seaweeds have very high productivity, absorb large amounts of N, P and $CO_2$, and produce a lot of $O_2$. If periodically planted and harvested, they uptake the nutrients from marine environments and assimilate N and P [7].

Hence, seaweed aqueous extracts application as biostimulants are part of the solution, being environmentally friendly and safe for human and animal health as they are a natural input. The use of seaweed extracts was initially explored for its promissory effects on ripening, fruit size and quantity increase and only subsequently for its helpful action against abiotic and biotic stressors. Furthermore, new research demonstrates that seaweed extracts might increase fruit quality even in stress-free conditions [8–10].

The term biostimulants refers to biological substances or microorganisms that, when applied to plants (via root drench, foliar spray, or both), can stimulate natural processes in the plant and are responsible for efficient plant nutrient use and growth, as well as increased tolerance to abiotic and biotic stresses, regarding their plant-beneficial nutrient content [11]. Algal extracts, when applied in plants, are considered biostimulants, rather than fertilizers since they enhance plant defense against pathogens and plant growth [11].

The chemical constitution of seaweeds includes complex polysaccharides, fatty acids, vitamins, phytohormones and mineral nutrients. Seaweed extracts from *Ascophyllum nodosum* (brown algae) and *Kappaphycus alvarezii* (red algae) are already widely used in available commercial products such as PROFERTIL (with 20% of *A. nodosum* content, by "ADP Fertilizantes") for agriculture application since they contribute to plant growth promotion (by improving water uptake and nutrients), plant tolerance to biotic and abiotic stresses (salinity, heat stress, water stress and nutrient deficiency) and, consequently, increase yields. Seaweed aqueous extracts also enhance macro- and micro-nutrient content, which consequently breaks down seed dormancy and enhance root growth, flowering, fruit taste and quality, and improve crop production [11,12].

Estuarine environments are susceptible to over-enrichment of nutrients and other pollutants derived from human activity such as agriculture (nitrate runoff) and sewage [13,14]. Mondego estuary is located on the Atlantic coast of Portugal (40°80′80 N, 8°50′ W), with a drainage area of 6670 km², and is influenced by a warm-temperate climate. The last 7 km, near the mouth, consists of two arms separated by a river island, the Murraceira, formed by the deposition of detrital materials transported by the river. Hydraulic circulation in the southern arm has been mostly dependent on tides and the freshwater input from the Pranto River, a small tributary. Discharges from the Pranto are controlled by a sluice and regulated in accordance with the water needs of rice fields in the Lower Mondego Valley. Due to the drainage of agricultural land, in addition to being N- enriched ($NO_3$-N, $NO_2$-N and $NH_4$-N), the Pranto water discharge contains several pollutants, including suspended solids and pesticides [15]. One of the most evident signs of nutrient enrichment in estuaries is the establishment of several opportunistic seaweeds in their waters [13], able to tolerate a salinity gradient and light attenuation and make efficient use of nutrient-rich brackish waters, affecting the structure of the community and its primary productivity [16]. These fast-growing seaweeds have emerged as an untapped resource that, with a circular economy approach, could be suitable as a feedstock supply for exploitation by the agricultural sector and industry.

Thus, the present study aimed to study the potential of the estuarine seaweed aqueous extracts of green algae *Ulva lactuca* (Sea lettuce), brown algae *Fucus ceranoides* (Estuary Wrack) and red algae *Gracilaria gracilis* (Slender Wart Weed), on kale (*Brassica oleracea* L.) seed germination and seedling development and to discern their stimulating potential

(promoting rapid germination and/or a more vigorous seedling development), that can be used as a sustainable option in agriculture to help attain better growth and production of kale crops.

## 2. Materials and Methods

### 2.1. Seed Material

Kale seeds (*Brassica oleracea* L.) col Lisa Bacalan Grande (Semillas Battle S.A., Barcelona, Spain) were selected, counted (300 seeds) and stored in empty Eppendorfs in the dark in a fridge at 4 °C (Candy SFD2460E) until use.

### 2.2. Collection and Preparation of Seaweed Biomass

Seaweed samples of estuarine algal species *Ulva lactuca* (Sea lettuce, Chlorophyta), *Fucus ceranoides* (Estuary Wrack, Phaeophyceae) and *Gracilaria gracilis* (Slender Wart Weed, Rhodophyta) were collected from Mondego estuary (Figure 1), Figueira da Foz, Portugal (Figure 2), on 27 March 2021. The seaweeds were handpicked and transported to the laboratory inside plastic bags in a thermal suitcase. After that, all samples were washed thoroughly with seawater to remove unwanted impurities, adhered sand particles (sediments) and epiphytes, and then quickly washed with distilled water to remove excess salt. After this process, all samples were dried in an air-forced oven (Raypa DAF-135, R. Espinar S.L., Barcelona, Spain) at 60 °C for 48 h, to facilitate the extraction process, increase yield and conserve the biomass for longer. Finally, the samples were stored inside vacuum plastic bags (to protect from humidity) in the dark at room temperature until use.

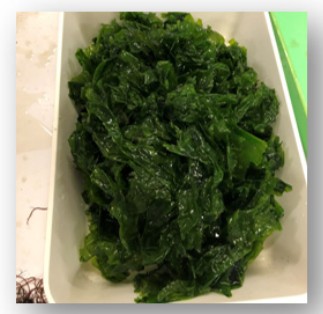 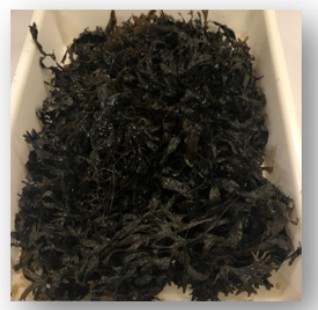 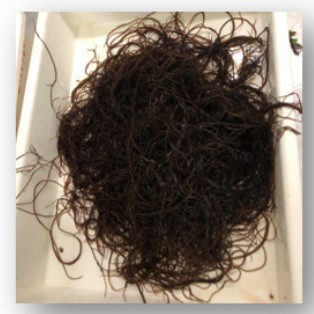

*Ulva lactuca*      *Fucus ceranoides*      *Gracilaria gracilis*

**Chlorophyta**      **Phaeophyceae**      **Rhodophyta**
Kingdom Plantae      Kingdom Chromista      Kingdom Plantae

**Figure 1.** Estuarine seaweeds *(U. lactuca, F. ceranoides* and *G. gracilis)* used in the aqueous extracts, and respective Phylum/ Class and Kingdom.

### 2.3. Macroelements and Physicochemical Content of Seaweeds

The determination of macro-elements and physicochemical content was made by chemical analysis by a credited external lab (Lusalgae). Moisture and ashes content was determined according to the international standard method 930.04 of the Official Methods of Analysis of AOAC International [17]. From the ashes obtained, the mineral and trace element content were analyzed through dry mineralization and assessed using flame atomic absorption spectrometry with the cathode corresponding to each element [18]. Crude Fiber and Total Carbohydrates were determined according to the standard method 930.10 of AOAC [17]. The total lipids content was quantified by GC/FID, using an SP-2560 capillary column per AOAC 2012.13 [17]. Total protein content was determined by the Dumas method per AOAC 968.06 [16]. Lead quantification was determined by Graphite Furnace Atomic Absorption Spectrometry (GFAAS) [19].

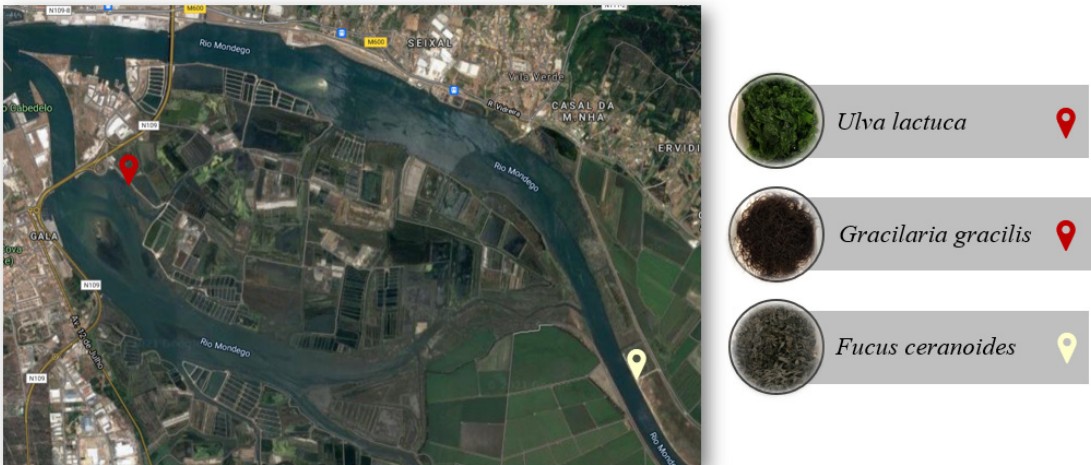

**Figure 2.** Location of seaweed sampling sites *(U. lactuca* and *G. gracilis*, harvested from the same site, and *F. ceranoides)* at Mondego estuary, Figueira da Foz, Portugal. Copyright: Google maps.

## 2.4. Preparation of Aqueous Seaweed Extracts (AEs)

All aqueous seaweed extracts were obtained according to the method described by Sousa et al. [20]. Briefly, dried seaweed samples were roughly cut into pieces and added to a blender (Moulinex LM811D) with distilled water, in a concentration of 0.12 g/mL, and blended for 5 min at 2500 rpm at room temperature. The viscous paste that resulted was filtered using a Büchner funnel with a piece of fabric above it connected to a Kitasato flask, under a vacuum. Larger residues, gathered by the fabric, were discarded and the crude liquid extracts (CE) were designated according to treatment and coded according to the genus and species: *U. lactuca* (UL), *F. ceranoides* (FC) and *G. gracilis* (GG) in sealed recipients kept at 4 °C in a fridge (Candy SFD2460E) for further analysis. Aqueous extracts (AEs) of each treatment (UL, FC, GG) were obtained from the respective seaweed stock of CEs and distilled water with a dilution of 1.43% [20,21].

Furthermore, pH, electrical conductivity (EC) and total dissolved solids (TDS) of the AEs were measured using a pH/Conductivity/TDS meter (Combo HI98129, HANNA instruments). All extracts were prepared and used on 17 May 2021.

## 2.5. Germination Assays

Kale seeds were treated with three different treatments (UL, FC, GG) and distilled water as control (C), in three replications (in 15 × 2 cm Petri dishes, each with 25 seeds). All seeds were disinfected in 2% sodium hypochlorite (José Manuel Gomes dos Santos, Portugal) (NaClO) for 1 min and washed in distilled water three times [22]. After that, the 25 seeds were placed inside sterilized Petri dishes, on a filter paper with cotton below, and 70 mL of AEs were added, as a respective seaweed treatment, except in the case of the control, where the same volume of distilled water was added. These 12 (three replications of the control and each treatment) Petri dishes were sealed with parafilm and placed in an incubator (Heraeus B5090E Incubator, Thermo Scientific, Osterode, Germany) at 23 °C and 29% relative humidity determined by a Nedis WEST100WT thermometer (Hertogenbosch, Netherlands) in the dark [20] for 7 days. To evaluate the seed germination, the variables analyzed were:

Germination percentage (GP), calculated by Equation (1), according to Hernández-Herrera et al. [23]:

$$GP = (\text{Number of germinated seeds/Total number of seeds}) \times 100 \qquad (1)$$

Seedling vigor index (SVI), determined using Equation (2), according to Orchard [24]:

$$SVI = (\text{Seedling length (cm)} \times \text{Germination percentage}) \qquad (2)$$

Measured variables also included apex length (measured from the cotyledon base to the apical bud) and radicle length (from the cotyledon to the tip of the longest root) measured using a ruler. The apex was separated from the radicle at the cotyledon base, and the apex fresh weight and fresh radicle weight were measured using an analytical scale (Kern, Germany).

### 2.6. Statistical Analysis

All data analyses were performed in Excel and subjected to the tests of significance using one-way analysis of variance (ANOVA) through Sigmaplot v.14.5 (statistical difference $p < 0.05$). A Bonferroni multiple comparison t-test was used after the rejection of the ANOVA null hypothesis to discriminate significant statistical differences between samples.

### 3. Results

#### 3.1. Physicochemical Content of AEs and Macroelements in Seaweeds

Carbohydrates, fiber and ash were the most abundant chemical components of green seaweed *U. lactuca* and lipid content was the lowest, compared to the other seaweeds. Red seaweed *G. gracilis* exhibited the highest values of protein and ash. Moisture content was relatively high in all three seaweeds (Table 1).

**Table 1.** Composition of seaweeds (%) used in the AEs preparation.

| Species | Moisture (%) | Protein (%) | Lipids (%) | Ash (%) | Fiber (%) | Carbohydrates (%) |
|---------|--------------|-------------|------------|---------|-----------|-------------------|
| *U. lactuca* | 94.8 ± 1.6 | 0.6 ± 0.1 | 0.1 ± 0.01 | 2.31 ± 0.11 | 1.5 ± 0.23 | 2 ± 0.1 |
| *F. ceranoides* | 96.7 ± 1.6 | 0.5 ± 0.1 | 0.4 ± 0.03 | 1.08 ± 0.05 | 0.6 ± 0.09 | 1 ± 0.1 |
| *G. gracilis* | 96.2 ± 1.6 | 0.9 ± 0.1 | <0.3 | 1.86 ± 0.09 | 0.4 ± 0.06 | 1 ± 0,1 |

The nutrient analysis revealed the macro (Na, K), trace element (Fe) and heavy metal ion (Pb) content in all samples (Table 2). The concentrations of Na and K were highest in the brown seaweed *F. ceranoides* and red seaweed *G. gracilis*, respectively. The concentration of Fe was highest in *G. gracilis*. The Pb concentration was lowest in *F. ceranoides* (Table 2).

**Table 2.** The content of macro, trace elements and heavy metal ions in seaweed species (mg/kg, dry wt.).

| Species | Na (mg/kg) | K (mg/kg) | Fe (mg/kg) | Pb (mg/kg) |
|---------|------------|-----------|------------|------------|
| *U. lactuca* | 0.41 ± 0.1 | 1589 ± 318 | 124 ± 25 | 3.4 ± 0.6 |
| *F. ceranoides* | 1.59 ± 0.32 | 3625 ± 725 | 92.3 ± 18.5 | <0.1 |
| *G. gracilis* | 0.67 ± 0.13 | 6841 ± 1368 | 521 ± 104 | 0.26 ± 0.05 |

#### 3.2. Characterization of Aqueous Seaweed Extracts AEs

Crude extracts (CE) of *U. lactuca* (UL), *F. ceranoides* (FC) and *G. gracilis* (GG) (Table 3) showed high EC and TDS values, so a dilution (1.43%) was performed to obtain suitable AEs (Table 4). Afterward, the pH of *F. ceranoides* and *G. gracilis* AEs were found to be slightly acidic compared to that of *U. lactuca* AE, which was neutral. The EC value was the lowest in the AE of *F. ceranoides*.

**Table 3.** Crude extracts (CEs) stock solution from seaweed samples observing their color and analyzing their pH from the, *U. lactuca* (UL), *F. ceranoides* (FC) and *Gracilaria gracilis* (GG) seaweed samples.

| Extract (CE) | Color | pH | TDS (mg/L) |
|--------------|-------|-----|------------|
| UL | Greenish yellow | 6.81 | >1997.71 |
| FC | Dark green | 6.66 | >1997.71 |
| GG | Brownish orange | 6.36 | >1997.71 |

**Table 4.** Aqueous extract (AE) analysis of pH, electrical conductivity (EC) and total dissolved solids (TDS) from the, *U. lactuca* (UL), *F. ceranoides* (FC) and *G. gracilis* (GG) seaweed samples.

| Extract (AE) | pH | EC ($\mu$S cm$^{-1}$) | TDS (mg/L) |
|---|---|---|---|
| UL | 7.07 | 327 | 164.81 |
| FC | 6.66 | 149 | 74.91 |
| GG | 6.36 | 288 | 143.83 |

*3.3. Effect of AEs on Kale Seed Germination*

Germination was observed 7 days after the experiment started in all treatments (Figure 3). The effect of the aqueous seaweed extracts of *U. lactuca* and *G. gracilis* presented an improved germination percentage (both 97,3%), in comparison with the control (96%) (Table 5). The treatment with *F. ceranoides* pointed to a lower germination percentage (94.7%).

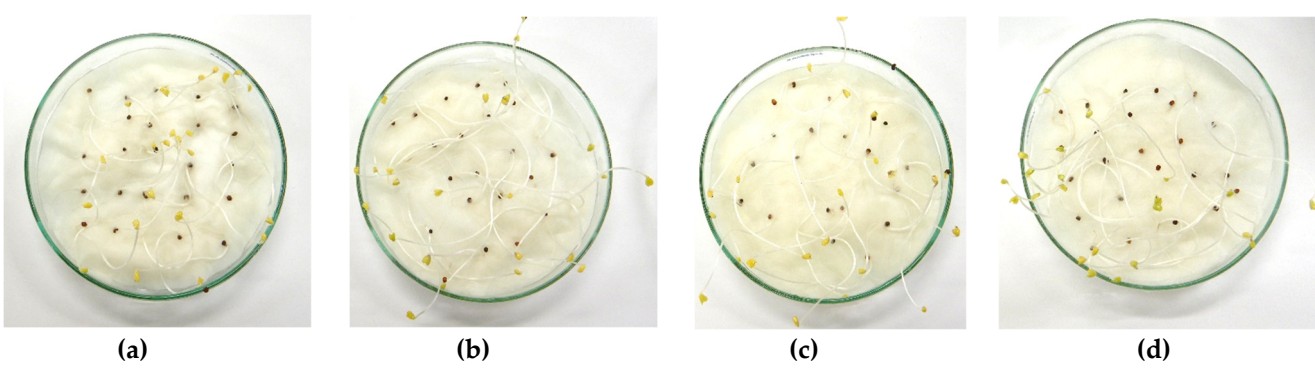

|     |     |     |     |
|:---:|:---:|:---:|:---:|
| (a) | (b) | (c) | (d) |

**Figure 3.** Photographic record of (**a**) control, (**b**) *Ulva lactuca*, (**c**) *Fucus ceranoides*, and (**d**) *Gracilaria gracilis*, after 7 days of incubation.

**Table 5.** Apex length (cm), radicle length (cm), average total length (cm), apex fresh weight (g), fresh radicle weight (g), average weight of seedlings, $\pm$ standard error (n = 3), germination percentage (GP) and seedling vigor index (SVI) of kale (*Brassica oleracea* L.) within the different treatments, Control (C), *U. lactuca* (UL), *F. ceranoides* (FC) and *G. gracilis* (GG).

| Measurements | C | UL | FC | GG |
|---|---|---|---|---|
| GP (%) | 96.0 $\pm$ 0.1 | 97.3 $\pm$ 0.1 | 94.7 $\pm$ 0.1 | 97.3 $\pm$ 0.1 |
| Apex Length (cm) | 6.77 $\pm$ 0.20 [a] | 7.69 $\pm$ 0.29 [a] | 7.14 $\pm$ 0.27 [a] | 9.03 $\pm$ 0.30 [a] |
| Radicle Length (cm) | 6.33 $\pm$ 0.34 [b] | 5.64 $\pm$ 0.30 [b] | 5.83 $\pm$ 0.32 [b] | 6.45 $\pm$ 0.30 [a] |
| Avg. total length (cm) | 13.10 $\pm$ 0.54 [a] | 13.33 $\pm$ 0.58 [a] | 12.97 $\pm$ 0.60 [a] | 15.48 $\pm$ 0.59 [a] |
| Apex Fresh Weight (g) | 0.052 $\pm$ 0.002 [b] | 0.065 $\pm$ 0.005 [b] | 0.054 $\pm$ 0.002 [b] | 0.069 $\pm$ 0.003 [a] |
| Fresh Radicle Weight (g) | 0.006 $\pm$ 0.001 [a] | 0.007 $\pm$ 0.001 [a] | 0.007 $\pm$ 0.001 [a] | 0.007 $\pm$ 0.001 [a] |
| Avg. weight of seedlings (g) | 0.059 $\pm$ 0.002 [a] | 0.071 $\pm$ 0.006 [a] | 0.062 $\pm$ 0.002 [a] | 0.076 $\pm$ 0.004 [a] |
| SVI | 1257.7 | 1297.5 | 1227.6 | 1506.9 |

[a, b] Values are average. Values with the same letter are not significantly different ($p > 0.05$).

*Gracilaria gracilis* treatment stimulated the longest apex length (9.03 $\pm$ 0.30 cm), but, statistically, there were no significant differences ($p > 0.05$), between the three treatments regarding apex length. The control and FC had the lowest apex lengths. When radicle length was measured, the *G. gracilis* treatment produced the best results (6.45 $\pm$ 0.30 cm) and was statistically different from the other treatments (FC and UL). The UL showed the lowest value (5.64 $\pm$ 0.30 cm) (Table 5).

For apex fresh weight parameters, the *G. gracilis* samples had the heaviest (0.069 $\pm$ 0.003 g), while the control was the lightest (0.052 $\pm$ 0.002 g). The *F. ceranoides* also showed a low value (0.054 $\pm$ 0.002 g), while *U. lactuca* showed an intermediate value (0.065 $\pm$ 0.005 g). Only the *G. gracilis* sample values were statistically significant (Table 5).

The fresh radicle weight was the same (0.007 ± 0.001 g) in all AE treatments, except in the control, where it was slightly lower (0.006 ± 0.001 g) without any statistical differences (Table 5).

Seeds treated with *U. lactuca* and *G. gracilis* AEs showed higher germination rates associated with greater seedling vigor index (SVI) (Table 5).

## 4. Discussion

Aqueous extracts are widely used and have proved to be very efficient, due to the ability to extract soluble compounds in water, such as sugars, polysaccharides, minerals, proteins, and other relevant compounds [25]. Thus, crude extracts appear to be the most efficient and low-cost utilization of seaweeds in the agricultural sector and could be effective as is seen with commercial seaweed biofertilizers. The waste (larger residues) that resulted from the production of these aqueous extracts also deserves attention for further studies, due to its suitability as a fertilizer as previously described by Sousa et al. [20].

The parameters of the AEs in this study, such as pH, electrical conductivity (EC) and TDS, affected kale seed germination. Changes in pH and EC in the extracts can affect seedling bioactivity [23].

pH affects seed germination and seedling development [21]. Germination was the best with pH = 7 since *B. oleracea* seedlings prefer neutral mediums. However, recent studies [26,27] show that, despite the idea that acidic substrates are detrimental to seed development, *B. olearacea* can benefit from a slightly acidic pH between 5.5 and 6.5. AEs of *F. ceranoides* and *G. gracilis* showed pH values slightly more acidic than that of the *U. lactuca* AE. According to Pacheco et al. [21], the slightly acidic AE of *G. gracilis* could be due to the high concentration of uronic acids present in this seaweed.

Most seeds are sensitive to salinity during germination and respective seedling growth. *B. oleracea* seedlings are moderately sensitive to NaCl, tolerating a maximum of 1000 $\mu S \cdot cm^{-1}$ [23,28]. In theory, a low value of EC is, in part, related to the absence of salts in the medium, increasing the chances of enhanced germination, as it allows seeds to imbibe water more efficiently. However, in this study, although all treatments presented low EC values of between 288 and 327 $\mu S \cdot cm^{-1}$, with slightly higher values in, *G. gracilis* and *U. lactuca*, respectively, they responded better in terms of germination rate and greater seedling vigor. During imbibition, the effect of salt is merely osmotic until a hydration threshold is surpassed [29]. Analogous studies, with even lower results, have been reported by Basher et al. [30], with seedling growth of tomato (*Lycopersicum* spp.), where the combination of water with seaweed treatments showed the highest germination rate with 27 $\mu S \cdot cm^{-1}$ with 0.1% seaweed concentration and lowest rate for 50 $\mu S \cdot cm^{-1}$ with 0.05% seaweed concentration, compared to the control [23].

Germination percentage and seedling vigor of *Brassica oleracea* improved when treated with *G. gracilis*. These results have been previously reported in several other experiments with *Gracilaria* in rice (*Oryza sativa*) that also revealed a greater germination percentage and seedling vigor [31].

*Brassica oleracea* seedling apex fresh weight and length improved, and the weight was statistically significant with *G. gracilis* treatment. A study conducted with tomato plants treated with extracts of red, brown and green seaweeds also generally resulted in increased plant height, leaf numbers, root width and root length, and an overall increase in biomass [12,32]. This could be due to growth-promoting chemicals present in the AE, such as cytokinin, auxins, gibberellins, abscisic acid (ABA) and ethylene [11]. Growth hormones are essential for increasing cell size and division and complement each other; for example, cytokinin is useful in shoot production and auxin in root development [11].

There was a very small radicular length and low fresh radicular weight improvement between the seedlings treated with all AEs except *G. gracilis*. A previous study made with polysaccharides extracted from *G. gracilis* also achieved relatively significant results in the radicular length of kale seedlings (*Brassica oleracea* L.) [21].

*Fucus ceranoides* treatment showed inhibitory results in all measured parameters, that this could be due to a large variety of factors, such as the percentage of solids (carbohydrates, minerals, and protein) present in the aqueous extracts influencing the rate of imbibition (movement of the water into the seed due to the difference in water potentials between the seed and its external liquid environment) [33]. Further study is needed with this seaweed to obtain more information.

Seaweed products exhibit growth-stimulating effects, and the use of seaweed as biostimulants in crop production is well established. According to data received from the external lab (Lusalgae), the three different seaweeds used for the extracts exhibited interesting biochemical parameters. The assay of the ash indicated the abundance of macro- and micro-elements. Generally, a significant promoting effect of seaweed liquid extracts is attributed to the presence of the macro elements (Ca, K, Na, P, Mg, N, S)and micronutrients (Cu, Zn, Co, Ni, Cl, Fe, Mn), vitamins and amino acids, in addition to protein and some growth-promoting substances as auxins, cytokinin and gibberellin, which influence cellular metabolism in supplemented seeds resulted in growth enhancement [21,34]. Polysaccharides being sugars are also known to improve plant growth in a similar way to hormones. Precursors of elicitor compounds can be present in the AEs, promoting germination and growth, and maintenance of plant health [30].

Further analyses of the macro- and micro-elements, amino acids, vitamins and phytohormones that affect cellular metabolism in seaweed-treated plants are needed to understand better the enhanced seed germination and seedling growth.

## 5. Conclusions

Seaweed extracts are considered environmentally friendly biofertilizers that help to mitigate the negative socioeconomic and environmental effects that synthetic fertilizers have. The current study supports the idea that traditional synthetic fertilizers can be applied in low quantities and, in part, be substituted by seaweed-originated biostimulants.

The seed germination study performed in this work with different treatments allowed us to define and discern the best seaweed aqueous extract source. It highlights the efficiency and use of the aqueous extracts (AEs), obtained from three seaweeds, *Ulva lactuca*, *Fucus ceranoides* and *Gracilaria gracilis*, as potential organic fertilizer. These AEs obtained from estuarine fast-growing seaweeds stimulated fast seed germination and posterior plant growth enhancement, providing a powerful and environmentally friendly approach for plant nutrient management in sustainable agriculture. Red algae, *Gracilaria gracilis*, extract stimulated the growth of kale seedlings, showing a raised germination percentage and seedling average total length. Hence, AEs from this seaweed are a potential candidate as an effective organic biostimulant when used alone or mixed with commercially available fertilizers.

Future studies can be carried out on the chemical and biochemical characterization of AEs, with an aim of better understanding their stimulating effect on the seeds, seedlings and, subsequently, plants.

This study also showed that seaweed aqueous extracts can, eventually, be used as an organic fertilizer in hydroponic plant growing systems. More study is needed concerning this.

**Author Contributions:** Conceptualization, M.M., J.C., D.P., K.B. and L.P.; Seaweed laboratory work, M.M., D.P. and J.C.; writing—original draft preparation, M.M., D.P. and J.C.; writing, review and editing, M.M., D.P., J.C., K.B. and L.P.; supervision, J.C., D.P., K.B. and L.P. All authors have read and agreed to the published version of the manuscript.

**Funding:** This work was financed by national funds through the FCT–Foundation for Science and Technology, I.P., within the scope of the projects UIDB/04292/2020–MARE–Marine and Environmental Sciences Centre and Associate Laboratory ARNET. João Cotas thanks to the European Regional Development Fund, through the Interreg Atlantic Area Program, under the project NASPA (EAPA_451/2016). This study had the support of national funds, through Fundação para a Ciência e Tecnologia (FCT), under the project LA/P/0069/2020, granted to the Associate Labora-tory ARNET.

**Institutional Review Board Statement:** Not applicable.

**Informed Consent Statement:** Not applicable.

**Data Availability Statement:** Data available from authors.

**Acknowledgments:** João Cotas thanks to the European Regional Development Fund through the Interreg Atlantic Area Program, under the project NASPA (EAPA_451/2016). This study had the support of national funds through the Fundação para a Ciência e Tecnologia (FCT), under the project LA/P/0069/2020 granted to the Associate Laboratory ARNET.

**Conflicts of Interest:** The authors declare no conflict of interest.

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
