# Peer review of "An experimental Study Investigating the Effects on Brassica oleracea: Estuarine Seaweeds as Biostimulants in Seedling Development?"

_phycology, doi:10.3390/phycology2040023_

Round 1

Reviewer 1 Report

In the attached file

Author Response

Reviewer 1:

I have carefully reviewed this manuscript. In this manuscript, the Authors examined biostimulant properties of three species of macroalgae – red, green and brown. This publication presents basic research, including general characteristics of the obtained extracts and preliminary tests on plants – germination tests. In the publication, it is worth adding what is the novelty of this work. There are many publications in the literature devoted to the use of water extracts from algae as plant growth biostimulants.

This paper is generally well written, using correct grammar and syntax, but there are many issues which require explanation/supplementation.

In my opinion, manuscript in the present form should be accepted after major revision.

Line 11: too long introduction, lack of information on the production of extracts and their composition;

Answer: Abstract was shortened.

Line 19: add in the bracket country to Mondego estuary;

Line 23: 15.482 cm – were the Authors able to measure the length of the seedlings with such accuracy? Please correct in the whole manuscript. Please add values for the control group;

Answer: Corrected.

Line 41: The authors devoted much attention to the nitrogen cycle in nature. I am missing more detailed information, what role can algae extracts play in this process?

Answer: Was added: “Seaweeds have an enormous potential for reducing global warming and climate change. Some cultivated seaweeds have very high productivity, absorb a large amounts of N, P, and CO2, and produce a lot of O2. If periodically planted and harvested, thus uptake the nutrients from marine environments and assimilate N and P [7].” Line 53-56

Line 111: I propose to delete “harvested from the same place” from the Figure caption;

Answer: Was deleted.

Line 122: should be “March”;

Line 126: “excretion process” or “extraction process”? Please check;

Line 127: delete dot after “oven”;

Line 135: please add the Author name – “described by ….”;

Answer: Corrected.

Line 140: “larger residues…were discarded” – what can be the application for the post-extraction residue? This may be important from the point of view of the circular economy approach to which the Authors referred in this publication. Please add explanation/proposal in the discussion;

Answer: In discussion section was added: “The waste (larger residues), that resulted from the production of these aqueous extracts, also deserves attention for further studies, since its ability as fertilizer as previously described by Sousa et al. [19].” Line 243-245

Line 148: dot before “All”;

Answer: Corrected.

Line 149: in my opinion, this section should be significantly improved – methodology for determining individual parameters should be added;

Answer: Methodology used was added on section 2.3. of materials and methods.

Line 179: parameters: EC and TDS are not presented in Table 2 for crude extracts, please add for comparison with Table 3;

Answer: Was added for comparison.

Line 184: were the measurements of pH, EC, TDS carried out in repetitions?

Answer: Yes and showed the same values.

Line 187: parts per million (ppm) is a jargon unit; instead of this please use mg/L;

Answer: Corrected as suggested.

Line 189: the order of the presented results is incorrect – at the beginning there should be the characteristics of the raw seaweeds (Table 4) and then the extracts obtained from it. The same parameters should be determined in seaweeds and extracts to check what has been extracted. More important is the chemical composition of the extract itself, since it will be applied to plants. Analyzes as in Table 4 should also be done for the extract;

Answer: The order was changed.

Line 194: please unify the units – differently in the title, differently in Table 4;

Answer: Corrected.

Line 200: it is not clear to me on what basis only Na, K, Fe and Pb were selected for analysis in seaweeds biomass? No analysis for primary (N, P, K) and secondary (S, Ca, Mg) nutrients and microelements. The multielement analysis should also be performed for seaweed extracts. Pb is typical heavy metal ion, not trace element

Answer: The analyses preformed on the external lab had the purpose for food industry, so that kind of data isn’t available.

Line 200: Table 5

- incorrect Table caption – g/100 g is not equal to mg/kg;

- not mg/Kg but mg/kg;

- for Pb should be 0.1 not 0,1;

- For Fe (UL) should be 124±25; please correct the number of decimal places in the whole manuscript (e.g., in Line 216 should be ±0.30, not 0.295);

Answer: Corrected.

Line 213: in my opinion, Figure 4 should be deleted, since it is repeated in the text (Line 205) and later in Table 6; please avoid repetitions;

Answer: Was deleted.

Line 220: between which exactly groups were there statistically significant differences – GG and FC, GG and UL? Please better explain

Answer: Was updated.

Line 223: delete “b” from ±0.002b;

Answer: Corrected.

Line 232: Table 6

- GP in Table 6 – was it possible for 0.096% of a seed to germinate? Please correct GP in this table, for example 96.0±1;

- standardize the number of significant digits and the number of digits after the decimal point in this table and in the whole work;

- results are presented as an average ± standard error; earlier in Figure 4 as a mean and standard deviation? Why such differences in the presentation of data? Please standardize – mean or average;

- the Authors used mean (average) to present your data – did you check the normality of distribution? It should be added to the section 2.6;

Answer: Corrected.

Line 238: I don’t agree with this statement – in this publication, only water extracts were produced, not with organic solvents. Additionally, the physicochemical composition of extracts was not performed;

Answer: This statement was improved to: “Aqueous extracts are widely used and proved to be the most efficient, due to the ability to extract soluble compounds in water, such as sugars, polysaccharides, minerals, proteins, and other relevant compounds [24]. Thus, crude extracts appear to be the most efficient, low-cost and could be effective such as the commercial seaweed biofertilizers.” Line 239-242

Line 262: please delete dot after [41];

Answer: Corrected.

Line 274: are there literature data that indicate the type of phytohormones that may be present in the tested seaweed? It's worth mentioning;

Answer: Yes, but only about the Phylum they belong to. I updated and mentioned some of them.

Line 277: please provide reference;

Line 290: “micro- and microelements”? Please correct;

Answer: Corrected.

Line 291: it is not clear for me why SLE abbreviation is introduced here…all abbreviations should be explained, but here instead of SLE you can use seaweed (algal) extracts;

Answer: Corrected to “seaweed liquid extracts”.

Line 292: it is not clear for me that from this list you only tested Na, K, Fe and Pb. There are many more important elements for plants that Na and Pb; Mn is not macro-, but microelement;

Answer: The analyses preformed on the external lab had the purpose for food industry, so that kind of data isn’t available.

Line 336: References

- publication titles in lowercase letters, e.g., Line 339, 346….;

- Latin names in italics, e.g., Line 406; 717….

Answer: Corrected

Reviewer 2 Report

An interesting article bringing new knowledge in the field of producing biostimulants from seaweed.

However, I have some questions for the authors regarding the chemical composition information of the extracts:

- why was such an insufficient amount of elements examined? Surely many more are present.

- doesn't the Pb content disqualify the extract?

- Isn't it worth testing the extract for microbiological purity at least according to EN ISO 6579-1:2017

There is no value in Figure 4. Without this information, the figure is redundant especially since I think the results from it are repeated in Table 6.

Please check the significance determinations of differences in Table 6 especially for Radicle Length and Avg. total length. 

Instead of using SE I suggest SD it allows for faster analysis.

The discussion was carried out correctly, however, the paucity of chemical composition studies of the extracts produced makes it impossible to relate most of the results of other researchers to the authors' own studies.

The authors, for example, cite literature in which the following were examined in the extracts: macro elements (Ca, K, Na, P, Mg, N, S, Mn) and micronutrients (Cu, Zn, Co, 292Ni, Cl, Fe), vitamins, amino acids, in addition to proteins and some growth-promoting substances as cytokinin, gibberellin, auxins. 

However, they themselves have not found such compounds and elements in their biostimulators.

I know that it is currently no longer possible to supplement with the above studies, so I recommend that the manuscript be accepted after minor revisions and the above comments be taken as a recommendation for future research.

Author Response

Reviewer 2:

The manuscript entitled “Estuarine seaweeds as biostimulants in seedling development” and authored by Madalena Mendes, Diana Pacheco, João Cotas, Kiril Bahcevandziev, and Leonel Pereira, deals with the investigation of the potential biostimulant effect derived from the application of aqueous extracts from seaweeds harvested in Mondego estuary, namely Gracilaria gracilis (Rhodophyta), Fucus ceranoides (Phaeophyceae) and Ulva lactuca (Chlorophyta), in kale (Brassica oleracea L.) seed germination and seedling development.

The manuscript contains information that can seriously contribute to knowledge in this field. It appears well written and structured, although several typos are present in the main text. However, I do not feel that this would be a problem that would compromise its publication in Phycology.

However, some revisions need to be made before I can consider this manuscript suitable for publication in the journal. Below is a series of comments, listed point by point:

  1. TITLE: I like the Idea of including a question as the title. However, it should be augmented by inserting a sentence that better describes the experimental study. Authors should consider adding a sentence such as "...... An experimental study investigating their effects on Brassica oleracea," or something similar.

Answer: Corrected as suggested to: “An experimental study investigating the effects on Brassica oleracea: Estuarine seaweeds as biostimulants in seedling development?”

  1. ABSTRACT: This section is too long. It should contain a maximum of 200 words. I also feel that the introduction section of the abstract section is excessively lengthy, The authors could shorten this section by reducing this part.

Answer: Abstract was shortened.

  1. KEYWORDS: The keywords should be completely changed. The utility of these terms is to facilitate the search of the article using common scientific search engines (PubMed, GoogleScholar, Scopus, etc.), which rely on the terms contained in title, abstract, and keywords. Consequently, using terms that are already in these sections as keywords is inappropriate. I strongly suggest that the keywords be changed before re-submission.

Answer: Keywords were updated to: Seaweeds aqueous extracts; Biostimulant; Seedling development; Brassica sp.

  1. INTRODUCTION: this section is too long and should be seriously shortened. In addition, some issues are present:
  2. The main focus of the authors' research is not Kale, but seaweed extracts used as biostimulants. Accordingly, lines 31-37 should be moved to another part of the introduction. This section should begin at line 38.

Answer: Corrected as suggested.

  1. Line 54-61: as correct as the concept is, it should be better explained, especially considering that the use of algae extracts as biostimulants against biotic stresses is a fairly recent discovery. The authors should particularly mention that the use of seaweed extracts was at first investigated for its beneficial effects on ripening, increase in fruit size and numbers (10.21608/jalexu.2021.105537.1019), and only later its beneficial action against abiotic (10.3390/agriculture11060557) and biotic (10.3390/md19020059) stresses was proven. In addition, emerging research also shows how seaweed extracts are able to improve fruit quality even under stress-free conditions (10.3390/biom10121662). I strongly encourage the authors to fix this part and include the suggested references since they are very recent and useful to the argument of their introduction.

Answer: Suggested references were added on introduction section at Line 55-59.

  1. Line 63-85 contiene informazioni non essenziali per una sezione introduttiva. Questa parte potrebbe essere rimossa al fine di rendere più corta questa sezione, o essere mossa nella parte di Discussione.

Answer: This part was shortened.

  1. In LINE 89, is not essential include commercial name of this formulation.
  2. Figure 1 should be moved to Material and Method section.

Answer: Corrected as suggested.

  1. MATERIALS AND METHODS: generally speaking, this section is well descriptive in all its parts, with the only problem being related to section 2.4. I fully understand the need to outsource some analyses, especially if they are not routine in the own laboratory. However, the methods must be equally and accurately described. Please include this information.

Answer: section 2.4. was updated.

  1. RESULTS: In Table 6, a standard deviation of 0 does not make any sense. Please fix this value.

Answer: Corrected.

Reviewer 3 Report

The manuscript entitled “Estuarine seaweeds as biostimulants in seedling development” and authored by Madalena Mendes, Diana Pacheco, João Cotas, Kiril Bahcevandziev, and Leonel Pereira, deals with the investigation of the potential biostimulant effect derived from the application of aqueous extracts from seaweeds harvested in Mondego estuary, namely Gracilaria gracilis (Rhod ophyta), Fucus ceranoides (Phaeophyceae) and Ulva lactuca (Chlorophyta), in kale (Brassica oleracea L.) seed germination and seedling development.

The manuscript contains information that can seriously contribute to knowledge in this field. It appears well written and structured, although several typos are present in the main text. However, I do not feel that this would be a problem that would compromise its publication in Phycology.

However, some revisions need to be made before I can consider this manuscript suitable for publication in the journal. Below is a series of comments, listed point by point:

1.       TITLE: I like the Idea of including a question as the title. However, it should be augmented by inserting a sentence that better describes the experimental study. Authors should consider adding a sentence such as "...... An experimental study investigating their effects on Brassica oleracea," or something similar.

2.       ABSTRACT: This section is too long. It should contain a maximum of 200 words. I also feel that the introduction section of the abstract section is excessively lengthy, The authors could shorten this section by reducing this part.

3.       KEYWORDS: The keywords should be completely changed. The utility of these terms is to facilitate the search of the article using common scientific search engines (PubMed, GoogleScholar, Scopus, etc.), which rely on the terms contained in title, abstract, and keywords. Consequently, using terms that are already in these sections as keywords is inappropriate. I strongly suggest that the keywords be changed before re-submission.

4.       INTRODUCTION: this section is too long and should be seriously shortened. In addition, some issues are present:

a.       The main focus of the authors' research is not Kale, but seaweed extracts used as biostimulants. Accordingly, lines 31-37 should be moved to another part of the introduction. This section should begin at line 38.

b.       Line 54-61: as correct as the concept is, it should be better explained, especially considering that the use of algae extracts as biostimulants against biotic stresses is a fairly recent discovery. The authors should particularly mention that the use of seaweed extracts was at first investigated for its beneficial effects on ripening, increase in fruit size and numbers (10.21608/jalexu.2021.105537.1019), and only later its beneficial action against abiotic (10.3390/agriculture11060557) and biotic (10.3390/md19020059) stresses was proven. In addition, emerging research also shows how seaweed extracts are able to improve fruit quality even under stress-free conditions (10.3390/biom10121662). I strongly encourage the authors to fix this part and include the suggested references since they are very recent and useful to the argument of their introduction.

c.       Line 63-85 contiene informazioni non essenziali per una sezione introduttiva. Questa parte potrebbe essere rimossa al fine di rendere più corta questa sezione, o essere mossa nella parte di Discussione.

d.       In LINE 89, is not essential include commercial name of this formulation.

e.       Figure 1 should be moved to Material and Method section.

5.       MATERIALS AND METHODS: generally speaking, this section is well descriptive in all its parts, with the only problem being related to section 2.4. I fully understand the need to outsource some analyses, especially if they are not routine in the own laboratory. However, the methods must be equally and accurately described. Please include this information.

6.       RESULTS: In Table 6, a standard deviation of 0 does not make any sense. Please fix this value.

Author Response

Reviewer 3:

An interesting article bringing new knowledge in the field of producing biostimulants from seaweed. 

However, I have some questions for the authors regarding the chemical composition information of the extracts:

- why was such an insufficient amount of elements examined? Surely many more are present.

Answer: The analyses preformed on the external lab had the purpose for food industry, so that kind of data isn’t available.

- doesn't the Pb content disqualify the extract?

- Isn't it worth testing the extract for microbiological purity at least according to EN ISO 6579-1:2017

There is no value in Figure 4. Without this information, the figure is redundant especially since I think the results from it are repeated in Table 6.

Answer: Figure 4 was deleted.

Please check the significance determinations of differences in Table 6 especially for Radicle Length and Avg. total length. 

Answer: Corrected as suggested.

Instead of using SE I suggest SD it allows for faster analysis.

Answer: Corrected as suggested.

The discussion was carried out correctly, however, the paucity of chemical composition studies of the extracts produced makes it impossible to relate most of the results of other researchers to the authors' own studies.

The authors, for example, cite literature in which the following were examined in the extracts: macro elements (Ca, K, Na, P, Mg, N, S, Mn) and micronutrients (Cu, Zn, Co, 292Ni, Cl, Fe), vitamins, amino acids, in addition to proteins and some growth-promoting substances as cytokinin, gibberellin, auxins. 

However, they themselves have not found such compounds and elements in their biostimulators.

I know that it is currently no longer possible to supplement with the above studies, so I recommend that the manuscript be accepted after minor revisions and the above comments be taken as a recommendation for future research.

Answer: Although currently is no longer possible to supplement with the mentioned studies, these comments will be taken in account for the future studies.

Round 2

Reviewer 1 Report

The revised version reads well. I am satisfied with the revisions, corrections, or additions, this manuscript can now be accepted for publication.

Reviewer 3 Report

The author followed and revised the manuscript as suggested. The manuscript is now suitable as publication.